# Examining Predictors of Psychological Well-Being among University Students: A Descriptive Comparative Study across Thailand and Singapore

**DOI:** 10.3390/ijerph20031875

**Published:** 2023-01-19

**Authors:** Wareerat Thanoi, Nopporn Vongsirimas, Yajai Sitthimongkol, Piyanee Klainin-Yobas

**Affiliations:** 1Department of Mental Health and Psychiatric Nursing, Faculty of Nursing, Mahidol University, Salaya 73170, Thailand; 2Alice Lee Centre for Nursing Studies, National University of Singapore, Singapore 117597, Singapore

**Keywords:** stress, resilience, mindfulness, psychological well-being, university students

## Abstract

Background: Psychological well-being (PWB) is a significant indicator of positive psychology. Thus far, the predictors of PWB are not well-understood among university students in Asian countries. Purpose: This study aimed to investigate the relationships between PWB and its predictors (stress, resilience, mindfulness, self-efficacy, and social support) in Thai and Singaporean undergraduates. Stress is perceived to have a negative influence on PWB, but mindfulness, resilience, self-efficacy, and social support indicate positive influences. Methods: A cross-sectional descriptive predictive research design was used with 966 Thai and 696 Singaporean university students. After calculating an adequate sample size and performing convenience sampling, we administered the following six standard scales: the Perceived Stress Scale, the Connor–Davidson Resilience Scale, the Mindfulness Awareness Scale, the General Self-Efficacy Scale, the Multi-dimensional Scale of Perceived Social Support, and the Psychological Well-being Scale, along with a demographic questionnaire. Descriptive statistics, correlation analysis, and structural equation modeling were performed for participants’ PWB. Results: Mindfulness had significant effects on both factors of PWB, including autonomy and growth, and cognitive triad, across two samples. In the Thai sample, resilience most strongly predicted autonomy and growth and perceived stress did so the cognitive triad, whereas in the Singaporean sample, perceived control most strongly predicted autonomy and growth and support from friends did so the cognitive triad. Conclusion: These findings provide specific knowledge towards enhancing psychosocial interventions and promoting PWB to strengthen mindfulness, resilience, perceived control of stress, and social support.

## 1. Introduction

The prevalence of mental health problems among university students is alarming, with depression (76%), anxiety (88.4%), stress (84.4%), and other mental disorders (45.5 %) [1,2,3] dominating. Such problems have an impact on students’ psychological well-being (PWB) [4]. While pursuing education in colleges, university students in Asian countries commonly experience academic stress that may also affect their PWB [5]. Students in Thailand feel under substantial pressure from their parents and teachers to demonstrate good performance in university and achieve excellent grades [6]. Among university students, academic pressure is higher because of high workload, insufficient support from faculty members, and an unsupportive university climate [6,7]. Furthermore, academic excellence plays a significant role in determining students’ career success; therefore, some university students feel more stress and experience mental health problems such as anxiety, depression, substance abuse, aggression, sleep problems, suicidal ideation, and other behavioral problems [6,8,9,10]. Assessing university students’ PWB or mental health is an important task for mental health professionals.

PWB is a significant indicator of positive psychology. PWB has been defined in two different ways: (a) subjective well-being (happiness, positive affect, and life satisfaction) [11] and (b) eudaemonic well-being, including autonomy, self-acceptance, purpose in life, personal growth, positive relations with others, and environmental mastery [12,13]. Prior to establishing prevention interventions, one must fully understand the relationships between PWB and related variables. Evidence has revealed that PWB is negatively correlated with depression [14,15] Furthermore, several studies have suggested that PWB is related to perceived stress, resilience, mindfulness, perceived self-efficacy, and social support among university students [10,16,17,18]. However, most of these studies were conducted in Western countries and emphasized negative variables (such as stress) rather than positive aspects (such as resilience and self-efficacy). According to the positive psychology domain, disease prevention, mental health promotion, positive emotions, and optimal functioning are tantamount to pathology, dysfunction, and treatments [19]. Therefore, exploring both positive and negative variables is essential.

Stress is conceptualized as a relationship between individuals and their environment, occurring when persons appraise a situation as a threat exceeding their available coping resources [20]. Stress is found to be an important variable influencing psychological problems [10,21,22] and PWB [23,24]. Among medical students, a randomized controlled trial (RCT) indicated that stress management and resilience training significantly improved resilience, perceived stress, anxiety, and quality of life at eight weeks post-intervention [24].

Self-efficacy is defined as individuals’ beliefs in their capabilities to produce designated levels of performance that exercise influence over events that affect their lives [25]. During the transition to higher education, university students might experience feelings of vulnerability and lack of control over their academic lives, affecting their self-efficacy. Nevertheless, studies have revealed that self-efficacy is positively related to PWB (the absence of psychological problems) [23,26,27]. Another study found that self-efficacy and resilience lead to academic success (an indicator of well-being) among baccalaureate nursing students [26].

Resilience refers to individuals’ ability to change and deal with stressful events and adversity [28]. It reflects positive environmental adaptation despite risky situations and difficulties [29] that vary depending on individuals’ context, age, gender, time, and original culture [28]. Resilience is related to emotional problems such as anxiety, depression, hopelessness [10,30], and PWB [31,32,33]. One study examined the predicting effect on resilience, stress, mindfulness, and self-efficacy in PWB among Australian undergraduate nursing students. It found resilience to be the strongest predictor [33].

Mindfulness, also called “cautious attention”, refers to individuals’ ability to self-regulate attention to the present moment and to orientate toward acceptance, openness, and curiosity [34]. Evidence indicates that mindfulness is a strong predictor of PWB [23,33,35,36]. A study on 76 experienced meditators suggested that practicing mindfulness was significantly correlated with psychological well-being [35].

Social support, an external protective factor, is defined as individuals’ perception of adequate and valuable support influencing adjustment [37]. Social support can be received from significant others, family members, friends, and teachers [38]. Studies have demonstrated social support as being significantly associated with PWB [23,32,39]. In the Philippines, Klainin-Yobas et al. [23] examined PWB’s predicting factors among university students and found that social support from family and friends significantly predicted positive individual PWB.

### The Current Study

Although previous studies have examined factors influencing PWB among undergraduate students [23,33], little is known about the magnitude of the relationships among those predicting factors. In addition, only a few studies have explored cross-cultural differences in PWB predictors, particularly in Asian countries. Therefore, this study examined associations among stress, self-efficacy, resilience, mindfulness, social support, and PWB in Thai and Singaporean undergraduates. The research questions are: (a) What are the patterns of relationships between stress, self-efficacy, resilience, mindfulness, social support, and PWB among undergraduate university students in Thailand? (b) What are the similarities and differences in the patterns of relationships between Thai and Singaporean samples?

This study was conducted at two government-supported universities in Thailand and Singapore. The rationales for using the two universities are outlined here. First, they are the top universities in each country, with high standards of learning environments, curriculums, faculty members, and various schools/faculties (such as Medicine, Science, Law, Art and Engineering). Second, students in both universities face high competition, and they must maintain high academic performance, possibly leading to academic stress. Third, both universities are located in a big city with a high cost of living. Students may experience additional stress concerning daily living (such as financial burden, traffic jams, among others). Finally, both universities provide mental health services, and our findings can be helpful to improve such services for students.

## 2. Materials and Methods

### 2.1. Research Design

To answer the research questions, this study used a cross-sectional descriptive predictive research design, which allowed us to examine the phenomena of interest in natural settings [40].

### 2.2. Participants and Setting

The target populations were Thai and Singaporean undergraduate university students, regardless of faculty/school, academic year, or sociological background. Potential participants were excluded if they were diagnosed with medical and mental illnesses by physicians and/or psychiatrists. Convenience sampling was employed to recruit potential participants from all faculties, categorized into six groups: (1) Environment and Natural Resources; (2) Linguistics, Culture, and Education; (3) Medical Science; (4) Public Health; (5) Science and Technology; and (6) Social Science, Humanities, and Liberal Arts.

A sample size was determined through online power analyses for structural equation modeling [41]. An effect size of 0.88 was calculated according to findings from a previous study investigating stress, self-efficacy, and PWB among nursing students [42]. Using the effect size of 0.88, with the number of latent variables = 6, the number of observed variables = 8, power = 80%, and significance level = 0.05; an appropriate sample size for this study was determined to be at least 589 participants [41].

### 2.3. Recruitment Procedure

In Thailand, the researchers initially contacted a dean of each faculty to ask permission to collect data in their respective faculties. Afterwards, we organized meetings with undergraduate students in each faculty to explain the study and request their participation. Interested students were asked to sign the consent form and complete the self-reported paper-and-pencil questionnaire on the spot. Alternatively, students could contact the researchers later if they needed more time to consider. Approximately 1000 students attended the meeting, and 966 agreed to participate. Given that the questionnaires were anonymous, we were unable to identify reasons for non-participation.

In Singapore, the researchers sought permission from the president of the university to collect data from undergraduate students. Afterwards, the researchers sent an invitation e-mail to undergraduate students using the university e-mail list. The e-mail contained information about the study, the researchers’ contact information, and a link to an online questionnaire. Interested students were requested to click on the link, sign the online consent form and complete the questionnaire. Approximately 5000 students were approached via e-mail and 673 completed the online questionnaire anonymously. Given the anonymity, we did not know who did/did not participate, and thus, we were unable to identify reasons for non-participation.

### 2.4. Ethical Consideration

In Thailand, this study received ethical approval from the Human Rights Committee Related to Human Experimentation before commencing [COA. No. 2014/059.0805]. In Singapore, we received ethical approval from the University Institutional Review Board [NUS-IRB Reference Code: 12-385E]. All participants in Thailand and Singapore were asked to sign a written consent form prior to participating in the study.

### 2.5. Measures

We collected data through anonymous self-reported questionnaires, encompassing demographic information and the following instruments: the Connor–Davidson Resilience Scale [28]; Perceived Stress Scale [43]; General Self-Efficacy Scale [44]; Multi-dimensional Scale of Perceived Social Support [45]; and Psychological Well-being Scale [12]. The demographic information included gender, age, nationality, religion, ethnic group, faculty and school, academic year, and annual family income.

In Thailand, we used the Thai-version, hard-copy, self-reported questionnaire, whereas the English-version and online questionnaires were administered in Singapore. There are two reasons for using different questionnaire formats. First, in Singapore, the university e-mail list of potential participants could be accessed by the researchers, who were also the university teachers. However, such an e-mail list was not accessible by the researchers in Thailand, and a face-to-face platform was required to meet the students. Second, Singaporean students were familiar with e-mail invitations and the online questionnaire, whereas Thai students were accustomed to hard copies. Furthermore, the response rate would have been very low if we had used the online version in Thailand.

### 2.6. Perceived Stress

The 10-item Perceived Stress Scale (PSS: [43]) was used to measure the degree of individuals’ thoughts and feelings about current events during the previous month. Each item was rated on a 5-item scale, 0 (never) to 4 (very often). The total score ranged from 0 to 40, with higher scores suggesting higher stress. Cronbach alphas were originally reported in the range of 0.84–0.86 among American graduate students [46]. With Thai adult participants, its test–retest reliability was 0.82 and the Cronbach alpha was 0.88 [47]. In this study, factor analyses revealed that PSS had two factors, perceived stress and perceived control, and the Cronbach alphas were 0.81 and 0.75 in Thailand, and 0.85 and 0.77 in Singapore.

### 2.7. Self-Efficacy

The Generalized Self-Efficacy Scale (GSES: [44]) consists of 10 items, which are rated on a 4-point scale ranging from 1 (not at all true) to 4 (exactly true). The Cronbach alphas of the GSES were originally reported in the range of 0.76–0.90 in adults and adolescents [44]. For the Thai version, the value was 0.84, suggesting good internal consistency [48]. In this study, GSES contained one factor. The Cronbach alphas in the Thai and Singaporean samples were 0.86 and 0.89, respectively.

### 2.8. Resilience

The 10-item Connor–Davidson Resilience Scale [49] was used to assess resilience by rating items on a 5-point scale from 0 (not true at all) to 4 (true all the time). Total scores vary from 0 to 40, with higher scores indicating higher resilience levels. Cronbach’s alpha was originally reported as 0.95 among American undergraduates [49]. Using the back-translation method, we translated and validated the Thai CD-RISC version. In this study, a factor analysis indicated that CD-RISC encompassed one structure, and the Cronbach alpha values of Thai and Singaporean university students were 0.86 [50] and 0.89, respectively, signifying good reliability.

### 2.9. Mindfulness

Mindfulness was measured by the 15-item Mindful Attention Awareness Scale (MAAS) [51], using a 6-point scale ranging from 1 (almost always) to 6 (almost never). Total scores range from 15 to 90, with higher scores reflecting higher levels of mindfulness. MAAS is psychometrically sound, given its good range of internal consistency across several samples (œ = 0.80–0.87) and excellent test–retest reliability over a 1-month period (r = 0.81) [51]. The Thai version of MAAS was validated with 385 Thai college students, and the results from a confirmatory factor analysis suggested a single-factor structure and the scale’s validity [52]. This study showed that MAAS had one factor for both samples, and the Cronbach alpha values were 0.88 and 0.97, respectively, indicating good reliability.

### 2.10. Social Support

The 12-item Multi-dimensional Scale of Perceived Social Support (MSPSS; [43]) was used to assess an individual’s perceived social support by rating items on a 7-point scale, from very strongly disagree (1) to very strongly agree (7). It entails three factors of support—friends, family, and significant others. Total scores range from 1 to 84, with higher scores signifying greater perceived social support. The MSPSS was originally tested on American university students, and the Cronbach alphas were in the range of 0.84–0.92 for total scores, 0.81–0.98 for the “family” subscale, 0.90–0.94 for the “friends” subscale, and 0.83–0.98 for the “significant others” subscale [45]. In 2005, Boonyamalik translated the Thai version of MSPSS with the back-translation method. It showed good reliability, with the Cronbach alpha in the range of 0.88–0.89 [53,54]. This study’s internal consistency regarding the two samples’ three factors was as follows: Cronbach’s alphas for social support from friends = 0.88; family = 0.90; and significant others = 0.91 for Thai university students. Those for Singaporean university students were 0.89, 0.92, and 0.86, respectively.

### 2.11. Psychological Well-Being

The 18-item Psychological Well-being Scale (PWBS) [12] was used to assess university students’ psychological well-being on a 6-point scale from (1) strongly disagree to (6) strongly agree. Possible scores range from 18–108, with higher scores signifying better Cronbach alphas that were originally reported in the range of 0.87–0.93, suggesting good reliability [12]. The researchers translated this measurement into Thai using the back-translation method. For Thai university students, Cronbach’s alpha was 0.80. In addition, the current study’s factor analyses revealed that PWB encompassed two factors: autonomy and growth, and the negative triad. Cronbach’s alphas were 0.85 and 0.70, respectively, in Thai; 0.85 and 0.56, respectively, in Singaporean university students [55].

### 2.12. Data Analysis

The data analysis was performed in two phases using IBM SPSS Statistics version 18.0. The first phase involved entering the data and checking entry accuracy. In Thailand, one researcher manually entered the questionnaire items to SPSS, while another researcher double-checked the entered data against the corresponding questionnaires. In Singapore, online questionnaires were digitally transferred to SPSS software. Next, the data cleaning began for both Thai and Singapore data sets by running the frequency of each study variable to ascertain that there were no out-of-range values. Where applicable, cross-tabulation (a feature in SPSS) was operated to confirm skip-and-fill questions were entered correctly. Descriptive statistics (i.e., frequency, mean, standard deviation, and graphical displays) was carried out to describe participants’ characteristics and study variables. The psychometric properties of all measurements were tested by a factor analysis and internal consistency reliability (Cronbach’s alpha).

The second phase tested the predictors of PWB via structural equation modeling (SEM) using AMOS software. Specifically, PWB factors (autonomy and growth, and negative triad) [56] were submitted to AMOS as dependent variables. Predictors included perceived stress, perceived control, mindfulness, resilience, and social support (support from friends, support from family, and support from significant others). The strength of the predictor would be determined by a standardized regression coefficient (β), and statistical significance was set as α = 0.05. An overall fit of the SEM model was determined by (a) confirmatory fit index (CFI), Tucker–Lewis Index (TLI), and incremental fit index (IFT) > 0.90; and (b) root mean square error of approximation (RMSEA) < 0.08 [57].

## 3. Results

### 3.1. Demographic Information and Study Variables

For the Thai sample, 966 students responded using the paper-and-pencil questionnaires. The average age was 20.21 (SD = 1.51), with females (67.30%, n = 650), males (31.80%, n = 307), and missing data (0.90%, n = 9). Most Thai students were Buddhist (94.50%, n = 913), followed by Christian (2.10%, n = 20), and Islamic (1.30%, n = 13) (Table 1).

For the Singaporean sample, 696 students completed the online questionnaires. The average age was 22.39 (SD = 5.18), with females (59.10%, n = 411), males (27.70%, n = 193), and missing data (13.20%, n = 92). Most were Christian (24.70%, n = 172), followed by Buddhist (18.20%, n = 127), Islamic (7.90%, n = 55), and others (49.20%, n = 342).

For both Thai and Singaporean students, all study variables—perceived stress, perceived control, resilience, self-efficacy, and mindfulness; all support the variables autonomy and growth, as well as negative triad—display approximately normal distribution (Table 2 and Table 3, respectively).

### 3.2. Predictors of Psychological Well-Being

#### 3.2.1. Thai Sample

Figure 1 displays predictors of PWB among Thai youths, with solid lines representing predictors achieving statistical significance and broken lines representing insignificant regression paths. The findings suggested that the hypothesized model had an adequate fit to the data, evidenced by Chi-square per degree of freedom (χ^2^/df) = 2.46, CFI = 0.90, TLI = 0.90, IFI = 0.90, RMSEA = 0.04, 90% confidence interval of RMSEA = 0.038, 0.041. Furthermore, resilience (β = 0.62, *p* < 0.001), perceived control (β = 0.29, *p* < 0.001), mindfulness (β = 0.17, *p* < 0.001), support from significant others (β = 0.17, *p* < 0.001), and support from family (β = 0.17, *p* < 0.001) significantly predicted the PWB autonomy and growth factor, with 60.90% of variance explained by all independent variables. Additionally, mindfulness (β = −0.24, *p* < 0.001), perceived stress (β = 0.32, *p* < 0.001), and support from family (β = 0.11, *p* = 0.03) significantly predicted the cognitive triad factor of PWB, with all independent variables explaining 31.30% of variance.

#### 3.2.2. Singaporean Sample

Figure 2 suggests that the hypothesized model displayed an acceptable fit, as evidenced by χ^2^/df = 2.14, CFI = 0.90, TLI = 0.90, IFI = 0.90, RMSEA = 0.04, and the 90% confidence interval of RMSEA = 0.039,.042. Note that these fit indices are comparable with those in the Thai sample. Furthermore, the autonomy and growth factor of PWB was significantly predicted by resilience (β = 0.29, *p* = 0.005), perceived stress (β = −0.15, *p* = 0.02), perceived control (β = 0.43, *p* < 0.001), mindfulness (β = 0.18, *p* < 0.001), support from friends (β = 0.11, *p* = 0.004), and support from family (β = 0.12, *p* = 0.001).

The cognitive triad factor was significantly predicted by resilience (β = −0.29, *p* = 0.006), perceived stress (β = 0.36, *p* < 0.001), mindfulness (β = −0.20, *p* < 0.001), and support from friends (β = −0.40, *p* < 0.001). All independent variables explained 65.90% and 69.90% of variance on the autonomy and growth, and the cognitive triad, respectively.

Altogether, the findings from both the Thai and Singapore samples demonstrated that mindfulness had significant effects on PWB factors. In the Thai sample, resilience most strongly predicted the autonomy and growth factor, while perceived stress most strongly predicted the cognitive triad factor. In the Singaporean sample, perceived control and support from friends most strongly predicted the autonomy and growth factor, and the cognitive triad factor.

## 4. Discussion

This study’s results demonstrated that Thai and Singaporean university students had a medium level of the autonomy and growth factor and negative triad factor of PWB. Negative triad reflects individuals’ negative perceptions toward self, other people, and their future [56]. Such perceptions included disappointing achievements, difficulty making and maintaining relationships with others, and little purpose in life. In the following variables, these students’ PWB models differed slightly: perceived stress, perceived control, resilience, social support from family, friends, and significant others. Other variables were similar, specifically mindfulness and self-efficacy.

In this study, stress was separated into two components: perceived stress and perceived control. In the Thai sample, perceived stress correlated negatively with the negative triad PWB factor, while the relationship between perceived control and the autonomy and growth of PWB was positive. In another study, individuals who perceived their stress as threatening or harmful indicated their potential to cause damage, thus provoking negative emotions. If they perceived stress as challenges that they could control with sufficient coping resources, they understood the potential rewards and ability to experience positive emotions [57]. Thai university students who perceived their stress as a life-threatening situation or as stressful life events might manifest the negative triad of PWB. In contrast, university students who interpreted their stressful life events positively and believed in their ability to control stress had greater autonomy and PWB growth.

However, the finding of perceived control revealed that university students could handle stress effectively and that individuals with effective coping strategies might have greater PWB, in congruence with the literature [58,59]. In comparison with Singaporean university students, the PWB model showed perceived stress as not only significantly negatively correlated with the autonomy and growth PWB, but also as significantly positively correlated with the cognitive triad PWB. Simultaneously, the relationships between perceived control and both PWB components were similar in the Thai sample. A previous study also supported that perceived stress had a negative correlation with positive PWB and a positive correlation with negative PWB [33]. Hence, promoting an intervention program of PWB in each sample should be considered its difference.

Mindfulness significantly predicted PWB among both samples, congruent with previous studies [23,33,60,61]. In addition, studies have suggested that mindfulness can reduce negative emotions: depression, rumination, stress, anxiety, somatization, aggression, and avoidance behavior [61]. Indeed, all previous studies indicated that mindfulness might reduce negative emotions and, correspondingly, increase PWB. Consistent with this study’s result showing a higher level of mindfulness, university students tended to mention higher autonomy and PWB growth and a lower negative triad PWB factor. This tendency is congruent with the literature describing “mindfulness”—conceptualized as promoting individuals’ well-being—as awareness of the present moment and non-judgment [50]. Individuals with elevated mindfulness are aware of their surroundings, thoughts, and feelings, without fixating or labeling things as good or bad [50]. Instead, they extend attitudes of curiosity, patience, and non-judgment towards distress because they better attend to the present, reduce rumination, have a greater ability to control their emotions and behaviors, and use more and better adaptive coping and management techniques to deal with undesirable stressors. All of these lead to greater PWB [62,63,64]. Importantly, these findings revealed that cultural differences between Thai and Singaporean students did not influence mindfulness in promoting PWB.

Resilience most strongly predicted the autonomy and growth PWB in the Thai sample and in both components of PWB in the Singaporean sample. Similarly, university students in Australia who possessed greater resilience reported higher levels of PWB [33]. It is possible that highly resilient individuals could better recover from adverse events and adjust to stressful situations [29,65]; resilience buffers them from the stress of life events, so they perceive stress as a challenge that helps them develop environmental mastery, positive relationships, growth, and self-determination [66]. Resilient university students could reappraise negative experiences as positive episodes [67], thus reducing the risk of maladaptive outcomes [68]. Resilient Thai university students could effectively develop their autonomy and growth to deal with stress, while Singaporean university students could develop both PWB components. Therefore, among university students, resilience seems well established in the literature as links to PWB.

In both study samples, self-efficacy had no significant effects on PWB, even though it might be enhanced by accomplishment, and well-being might be enhanced by beliefs about capabilities [69]. These findings were incongruent with a previous study [23]. Following previous studies [23,33,70], support by family, friends, and significant others related meaningfully to PWB, indicating that social support could enhance a person’s ability to handle stress effectively and promote PWB [71]. Thai university students could apply perceived support from family and significant others to their autonomy and growth PWB and to the reduction in the PWB negative triad. Family social support essentially contributed to PWB in Thai university students because although some had moved away to study, family connectedness was still profound.

According to the self-determination theory (SDT), adolescents perceiving their parents as a supportive resource could develop their autonomy, including their natural desire to experience a sense of personal decision making, volition, and psychological freedom [72]. For Singaporean students, support from friends contributed to both PWB components, while support from family only promoted PWB autonomy and growth. Especially because of the competitive, international environment at the Singaporean university, perceived support from friends influenced PWB, and this result corresponded with a previous study of Filipino university students [23]. These findings support cultural differences between Thai and Singaporean students as influences in their daily living.

Finally, the findings showed slight differences between Thai and Singaporean students because of diverse cultural and academic environments. For instance, most university students in Singapore had come from foreign countries and manifested a wide range of skills and abilities; thus, international competition and high education qualifications became inherently tense for them. Moreover, Singapore’s average cost of living is quite high and leads university students to strive diligently for the highest-paying careers and to become financially independent.

Like most studies, this one had some limitations. First, the cross-sectional research had limited time to provide a deep understanding of individuals’ PWB development. Therefore, longitudinal research is needed in future studies. Second, both hard-copy and online self-reported questionnaires are considered subjective data and might be subject to social desirability. For a more accurate reflection of PWB, longitudinal research should be implemented. Third, the use of different formats of questionnaires might minimize the comparability of findings across the two samples. Finally, the use of convenient sampling might limit the generalizability of the research findings. Nevertheless, a large sample size in the two samples might minimize this issue.

The study’s results further suggested several important factors for recommendations. To improve university students’ PWB, implementing intervention programs—for example, mindfulness-based stress-reduction programs, resilience programs, and social support programs as part of university policy—would promote PWB and help prevent students’ mental health problems. Of course, the effectiveness of such intervention programs—concentrating on mindfulness, resilience, perceived control of stress, and social support—should be carefully and regularly evaluated.

## 5. Conclusions

This study compared predictors of PWB across the two samples. Both the Thai and Singaporean samples’ mindfulness had significant effects for both PWB factors. In the Thai sample, resilience most strongly predicted the autonomy and growth PWB, and perceived stress did so the cognitive triad PWB. In the Singaporean sample, perceived control most strongly predicted the autonomy and growth PWB, and support from friends did so the cognitive triad PWB. Future research should test this hypothesized model in other university samples and implement effective intervention programs to enhance PWB in undergraduate university students.

## Figures and Tables

**Figure 1 ijerph-20-01875-f001:**
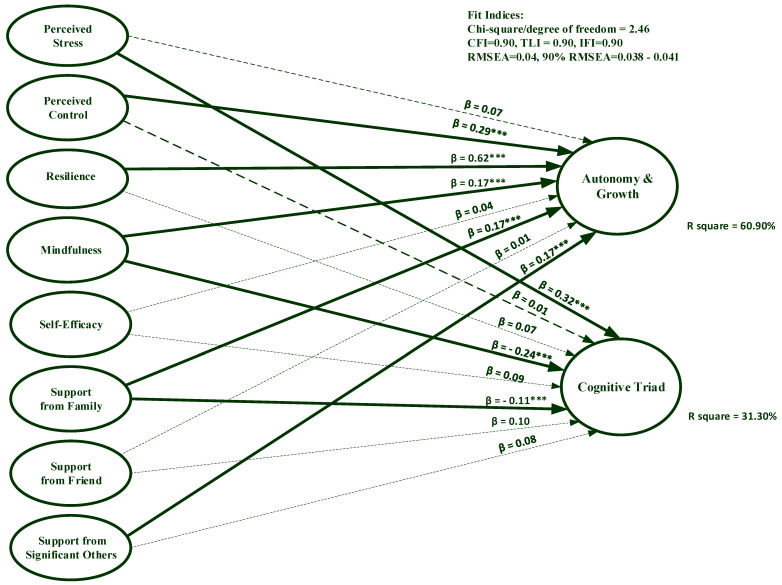
Predictors of psychological well-being among university students in Thailand. *** Significant at 0.001.

**Figure 2 ijerph-20-01875-f002:**
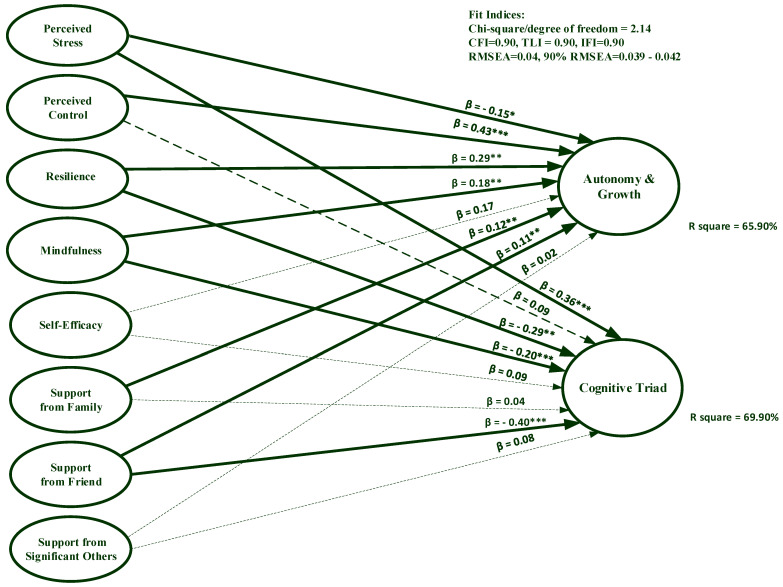
Predictors of psychological well-being among university students in Singapore. * Significant at 0.05, ** significant at 0.01, *** significant at 0.001.

**Table 1 ijerph-20-01875-t001:** Demographic of sample.

	Thailand (n = 966)		Singapore (n = 696)	
	n	%	n	%
Gender				
- Male	307	31.80	193	27.70
- Female	650	67.30	411	59.10
- Missing	9	0.90	92	13.20
Religion				
- Buddhism	913	94.50	127	18.20
- Christian	20	2.10	172	24.70
- Islam	13	1.30	55	7.90
- Others	-	-	342	49.20
- Missing	20	2.1		
School/Faculty				
- Liberal Arts	100	10.40	0	0
- Dentistry	91	9.40	0	0
- Engineering	110	11.40	0	0
- Medicine	48	5.00	0	0
- Nursing	211	21.80	86	12.4
- Science/Social Science	99	10.20	610	87.6
- Pharmacy	85	8.80	0	0
- Public Health	100	10.40	0	0
- Information Communication and Technology	70	7.20	0	0
- Music	43	4.50	0	0
- Missing	9	0.90	0	0
Study Year				
- Year 1	179	18.50	185	26.6
- Year 2	250	25.90	29	4.2
- Year 3	244	25.30	60	8.6
- Year 4	242	25.10	44	6.3
- Year 5	9	0.90	-	-
- Year 6	27	2.80	54	7.8
- Missing	15	1.60	372	46.50
	Mean	SD	Mean	SD
Age	20.21	1.51	22.39	5.18

**Table 2 ijerph-20-01875-t002:** Description of study variables for Thai sample (n = 966).

	Minimum	Maximum	Mean	Standard Deviation	Skewness	Kurtosis	Cronbach’s Alpha
Perceived stress	0	22	11.21	3.67	−0.00	0.32	0.81
perceived control	0	16	6.19	2.24	0.29	1.31	0.75
Resilience	7	40	27.69	5.17	−0.41	0.74	0.86
Self-efficacy	12	40	27.51	4.25	−0.02	0.44	0.86
Mindfulness	24	89	62.33	10.92	−0.15	−0.16	0.88
Support from family	6	28	23.64	4.32	−1.24	1.29	0.89
Support from friends	5	28	21.86	4.30	−0.82	0.69	0.91
Support from others	4	28	21.74	5.13	−0.92	0.68	0.91
Autonomy and growth of PWB	17	60	43.63	6.58	−0.66	0.91	0.85
Negative triad factor of PWB	6	36	25.25	4.637	−0.23	0.30	0.72

Note: PWB = psychological well-being.

**Table 3 ijerph-20-01875-t003:** Description of study variables for Singaporean sample (n = 673).

	Minimum	Maximum	Mean	Standard Deviation	Skewness	Kurtosis	Cronbach’s Alpha
Perceived stress	0	24	11.93	4.12	0.12	0.40	0.85
Perceived control	0	16	6.88	2.43	−0.03	0.84	0.77
Resilience	14	50	35.30	5.89	−0.04	0.49	0.89
Self-efficacy	10	40	29.30	3.81	−0.23	2.37	0.89
Mindfulness	17	89	61.62	10.13	−0.29	0.74	0.87
Support from family	4	28	20.05	5.05	−0.79	0.72	0.89
Support from friends	4	28	21.16	4.43	−1.29	2.85	0.92
Support from others	4	28	20.25	6.02	−0.73	−0.00	0.96
Autonomy and growth of PWB	9	54	40.32	6.37	−0.64	1.64	0.83
Negative triad of PWB	8	34	22.27	4.43	−0.29	0.02	0.56

Note: PWB = psychological well-being.

## Data Availability

Data is unavailable due to privacy and ethical restrictions from Institutional Review Board (IRB) of Mahidol University, Thailand.

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
