# Peer review of "Examining Predictors of Psychological Well-Being among University Students: A Descriptive Comparative Study across Thailand and Singapore"

_ijerph, 2023, doi:10.3390/ijerph20031875_

Round 1

Reviewer 1 Report

The manuscript addresses an interesting topic and the study seems to have been well planned and executed. However, there are a few noteworthy problems:

1.       The title could inform which samples are compared, instead of just saying that it is a "comparative study across two samples"

2.       The abstract must be improved.

·         I suggest that authors review the instructions for authors.

·         The abstract must begin with the background "Background: Place the question addressed in a broad context and highlight the purpose of the study" (IJERPH website).

·         Some words are in bold.

·         It is unnecessary to mention "After calculating an adequate sample size...".

3.       Introduction:

·         I see no need to address adolescence. Authors must focus on addressing university life, presenting arguments for the relevance of their study. I do not understand why the authors included the subheading "background" in the introduction section.

·         "The introduction should briefly place the study in a broad context and highlight why it is important. It should define the purpose of the work and its significance, including specific hypotheses being tested. The current state of the research field should be reviewed carefully and key publications cited" (IJERPH website).

4.       Materials and Methods:

·         Participants: Authors must inform how they reach the number of participants (how many were contacted? how many agreed to participate? how many were excluded? and why?).

·         The subsection "procedure" basically describes ethical procedures. Authors can summarize this information in a single sentence, which can be included at the end of the Materials and Methods section: "This study was approved by the Committee on Human Rights Related to Human Experimentation." If at each university the committee has a different name, the correct names must be presented. Regarding participants' consent, simply say "After obtaining informed consent from participants" and then explain how they responded to the questionnaire (e.g., online).

·         Do the authors use the snowball sampling method? The description suggests so.

·         The subsection “measures” can be more objective, with a simple phrase such as “The survey questionnaire included questions on sociodemographic characteristics and the following instruments:” (and then the instruments are presented as the authors did).

·         When mentioning the psychometric data of the instruments, the authors referred to American tests/ English versions. The ideal is to cite the original validation studies of these instruments.

·         It would be important for the authors to better explain how they conducted the analyzes to identify PWB predictors.

5.       Why are some words in bold?

I hope my comments may be a useful guide to improve the manuscript.

Kind regards.

Author Response

We are very grateful to the reviewers for their insightful comments on our paper. We have been able to incorporate changes to reflect all of the suggestions provided by the reviewers. We have highlighted the changes within the manuscript. Here is a point-by-point response to the reviewers' comments and concern. We also attached our response table in the attached file. 

Reviewer 2 Report

The manuscript by Thanoi et al., Examining predictors of psychological well-being among university students: A descriptive comparative study across two samples describes the use of an identical experimental structure, to investigate the effects of social support, stress, resilience, mindfulness, and self-efficacy on psychological well-being (PWB) in Thai and Singaporean undergraduates. The authors present evidence for the effects of the different factors amongst the two groups of students and the methods are well described and align with the research questions. The manuscript is stepwise structured (though some sections need to be looked at again), and the experiments are carefully designed to match the conclusions that are given.

I have carefully read through the manuscript and have these few comments to make to improve on understanding and add clarity.

1.     The paper does not list the affiliations of the authors, not emails

2.     Since the work focuses on the effects of mental problems on PWB, it will be important from the onset to define PWB and also include a general overview of the epidemiology of mental disorders

3.     Concerning the structure, I am uncertain about why there is a “Background” section added after the “Introduction”.

4.     I am interested in knowing the age ranges for the different cohorts of students and if all the students interviewed had reached the age of consent.

5.     It will be important to indicate why two different formats of the questionnaire were used at the different institutions and discuss the potential implications this could have on the obtained results.

6.     It will be important to include the reference for the ethical clearance document obtained from the study at the Singaporean university.

7.     Lastly, the paper has grammatical errors in so many places which sometimes obscure the meaning and makes understanding difficult. It will be important for the authors to do a proper proofreading and if possible, make use of a proofreading service (e.g., lines 125-126).

Author Response

We are greatful to your insightful comment on our paper. We have been able to incorporate changes to reflect all of the suggestions provided by you. We have highted the changes within the manuscript which suggested by the reviewers. Here is a point-by-point response to the comments and concern. 

Reviewer 3 Report

Thank you for the opportunity to review this paper. 

Overall, I think the article is well written. While the topic of factors associated with psychological well-being among university students is important for all stakeholders, my biggest concern for this paper is using a convenience sample, which impairs the findings' generalizability.

Overall, the study contributes moderately to the literature. Although it is well established that these factors are associated with psychological well-being, this paper provides information for a specific population that has not been well investigated.

Additionally, the authors need to recheck the in-text citations carefully. For example, in lines 44-46, it's better to add in-text citations.

Here are my specific comments:

Introduction:

·      I am curious why the authors did not provide the names of the universities. Are there specific concerns?

·      Please provide adequate information about the rationale and significance of why you chose Thailand and Singapore to compare. Did you intend to generalize the findings to ASEAN countries?

Methods:

·      Could you please explain why you chose two approaches (paper-based vs. online) to conduct the survey?

·      I suppose students in Thailand used the Thai version of the questionnaire. Did you use the English version in Singapore? 

·      Any quality assurance during the data collection and inputting process? Providing this detailed information will make readers confident about the data integrity and increase transparency for the study.

·      Given the potential broad audience, please define “negative triad”.

Results:

·      Please consider adding a table to summarize the demographic information of the study population. If possible, please also include the participants’ discipline (or faculty/college) and year in the university.

Discussion:

·      Please consider adding convenience sampling as one of the potential limitations.

·      Please consider adding the two formats (paper-based vs. online) in the two countries as one of the potential limitations.

·      Please consider adding the potential generalizability issues in your limitation section.

Author Response

We are grateful to the reviewers for their insightful comments on our paper. We have been able to incorporate changes to reflect most of the suggestions provided by the reviewers. We have highlighted the changes within the manuscript. Here is a point-by-point response to the reviewers' comments and concerns.

Round 2

Reviewer 1 Report

Dear authors,

Congratulations on improving the manuscript.You have addressed all my concerns. So that, I accept the publication of this manuscript.

Kind regards,
Reviewer